# Versatile Fluorine-Containing Building Blocks: *β*-CF_3_-1,3-enynes

**DOI:** 10.3390/molecules27249020

**Published:** 2022-12-17

**Authors:** Mingqing Liu, Zongxiang Yu, Jingtong Li, Yuanjing Xiao

**Affiliations:** Department of Chemistry, School of Chemistry and Molecular Engineering, East China Normal University, 500 Dongchuan Road, Shanghai 200241, China

**Keywords:** organofluorine compounds, fluorine-containing building blocks, *β*-CF_3_-1,3-enynes, diversity-oriented synthesis (DOS)

## Abstract

The development of diversity-oriented synthesis based on fluorine-containing building blocks has been one of the hot research fields in fluorine chemistry. *β*-CF_3_-1,3-enynes, as one type of fluorine-containing building blocks, have attracted more attention in the last few years due to their distinct reactivity. Numerous value-added trifluoromethylated or non-fluorinated compounds which have biologically relevant structural motifs, such as *O*-, *N*-, and *S*-heterocycles, carboncycles, fused polycycles, and multifunctionalized allenes were synthesized from these fluorine-containing building blocks. This review summarizes the most significant developments in the area of synthesis of organofluorine compounds based on *β*-CF_3_-1,3-enynes, providing a detailed overview of the current state of the art.

## 1. Introduction

Increasing demands for organofluorine compounds in human life [1,2,3,4,5,6,7,8,9,10,11,12] prompt chemists to develop many ingenious strategies to introduce the fluorine element into organic compounds [13,14,15,16,17,18,19,20,21,22,23,24,25,26,27]. Among these strategies, synthesis with fluorine containing building blocks is a very important method for the introduction of fluorine atoms or fluoroalkyl groups into target molecules [28,29,30,31,32,33,34,35,36]. *β*-CF_3_-1,3-enynes as one type of fluorine-containing building blocks have attracted more attention in the last few years due to their distinct reactivity. Numerous value-added fluorine-containing compounds which have biologically relevant structural motifs, such as *O*-, *N*-, and *S*-heterocycles, carboncycles, fused polycycles, and multifunctionalized allenes, were synthesized from these fluorine-containing building blocks. With the strong electron-negative effect of CF_3_-group on the C–C double bond, the electron density of conjugated enynes significantly changed, making the molecules exhibit distinct reactivity which enables diversity-oriented synthesis of organofluorine compounds. Within the remit of this review, *β*-CF_3_-1,3-enynes and their derivatives will all be discussed. The content summarized in this review are organized on the basis of the type of trifluoromethylated compounds obtained by using *β*-CF_3_-1,3-enynes and their derivatives as fluorine-containing building blocks, i.e., construction of trifluoromethylated carboncycles, construction of trifluoromethylated heterocycles and further subdivided according to the size of the ring, i.e., three-membered trifluoromethylated carboncycles, five-membered trifluoromethylated carboncycles, six-membered trifluoromethylated carboncycles, five-membered trifluoromethylated heterocycles, six-membered trifluoromethylated heterocycles and other value-added trifluoromethylated or non-fluorinated organic compounds.

## 2. Construction of Trifluoromethylated Carboncycles

### 2.1. Construction of Three-Membered Trifluoromethylated Carboncycles

In 2019, Wang and Liu reported highly diastereoselective cyclopropanation reactions of *β*-CF_3_-1,3-enynes with sulfur ylides via a maneuverable one-pot, two-step procedure. *β*-CF_3_-1,3-enynes undergo cyclopropanation reactions with sulfur ylides under mild reaction conditions without fluoride elimination, which affords the *cis*-isomer mainly. Interestingly, a sequential TBAF-mediated deprotection of the triisopropylsilyl group results in a diastereoenriched epimerization which gives rise to the transcyclopropanes as the sole isomers (Figure 1a) [37]. A base-triggered thermodynamic epimerization took place during the process, resulting in stereoselectivity enrichment (Figure 1b). This newly developed protocol was then applied to 2-CF_3_-3,5-diyne-1-enes for synthesis of diverse 1,3-diyne-tethered cyclopropanes [38].

Recently, a highly efficient solvent-controlled synthesis of bis(trifluoromethyl) cyclopropanes and bis(trifluoromethyl)pyrazolines via a [2 + 1] or [3 + 2] cycloaddition reaction of *β*-CF_3_-1,3-enynes with CF_3_CHN_2_ was developed by Cao and co-workers. The distribution of cyclopropanes and pyrazolines is remarkably dependent on the polarity of the solvent used. Less polar solvents, such as DCE, were suitable for the [2 + 1] cycloaddition reaction, whereas polar solvents, such as DMAc, were found to favor the [3 + 2] cycloaddition reaction [39] (Figure 2).

### 2.2. Construction of Five-Membered Trifluoromethylated Carboncycles

In 2011, Jeong and coworkers reported Pd-catalyzed intramolecular carbocyclization of 2-trifluoromethyl-1,1-diphenyl-1,3-enynes to afford 2-trifluoromethyl-1-methylene-3-phenylindene derivatives via electrophilic hydroarylation by the use of 10 mol% Pd(OAc)_2_ in the presence of CF_3_CO_2_H and CH_2_Cl_2_. It was postulated that this reaction proceeds via ortho-palladation of enynes to give a corresponding intermediate, which undergoes the insertion to a triple bond to give the vinylpalladium species. Protiodepalladation of vinylpalladium species affords 1-methylene indenes. The substrates were synthesized from pentafluoroethyl phenyl dithioketal in several steps. Both aryl and alkyl-substituted 1,3-enynes are tolerated, however, the stereoisomer ratios of 5/4 of 1-methylene indenes were afforded in this *5-exo-dig* carbocyclization. In the case of carbocyclization of trimethylsilyl- or triisopropylsilyl-substituted 1,3-enynes, reduced product was obtained [40] (Figure 3).

In an elegant piece of work, The Trost group reported palladium-catalyzed trimethylenemethane cycloadditions with α-trifluoromethyl-styrenes, trifluoromethyl-enynes, and dienes under mild reaction conditions. The trifluoromethyl group serves as a unique σ-electron-withdrawing group for the activation of the olefin toward the cycloaddition. This method allows for the formation of exomethylene cyclopentanes bearing a quaternary center substituted by the trifluoromethyl group (Figure 4) [41]. Diaminophosphite ligand was employed in this reaction to afford desired cycloaddition product in moderated to excellent yields (45–93%). The obtention of the cycloadduct unaccompanied by fluoride elimination may be suggestive of a concerted mechanism.

In 2017, we developed a concise approach to access ring-trifluoromethylated cyclopentene frameworks, utilizing silver-catalyzed double hydrocarbonation reaction of *β*-CF_3_-1,3-enynes with bisnucleophiles 1,3-dicarbonyl compounds [42] (Figure 5). *β*-CF_3_-1,3-enynes possessing electron-withdrawing aryl groups on the alkyne moiety smoothly underwent the cyclization reaction with 1,3-dicarbonyl compounds to give ring-trifluoromethylated cyclopentene in moderate to excellent yields. A series of *β*-diketones, malonate derivatives as well as *β*-ketoesters could be employed as reaction partner to afford the corresponding ring-trifluoromethylated cyclopentenes in good to excellent yields. Unsymmetrical 1,3-diketone or *β*-keto esters afford the desired cyclopentenes as a mixture of two diastereomers in excellent yields with moderate to good diastereoselectivity. The use of organic base effectively suppressed the defluorination process.

Interestingly, Chang and Zhang recently revealed that inorganic base K_3_PO_4_ can effectively promote synthesis of CF_3_-substituted cyclopentenes when using malononitrile as carbon nucleophile to react with *β*-CF_3_-1,3-enynes. Their developed protocol disclosed that the fluorine retention of *β*-CF_3_-1,3-enynes did not rely on organic base and the additional patterns did not depend on electron-deficient *β*-CF_3_-1,3-enynes. The reasons may mainly be due to the stability of in situ formed carbon anion intermediates containing the cyano group and CF_3_-substituted cyclopentenes with double bond migration formed [43] (Figure 6).

In 2020, Liu’s group reported the phosphine-catalyzed [3 + 2] cycloadditions of trifluoromethyl enynes/enediynes with allenoates to form cyclopentenes containing a CF_3_-substituted quaternary carbon center with great regioselectivity [44] (Figure 7). This reaction occurs with excellent regioselectivity under mild conditions, affording alkyne- and diyne-tethered cyclopentene derivatives containing a CF_3_-substituted quaternary carbon center in moderate to good yields.

### 2.3. Construction of Six-Membered Trifluoromethylated Carboncycles

In 2013, the Gevorgyan group demonstrated the synthesis of trifluoromethylated benzene derivatives via chemo- and regioselective Pd-catalyzed [4 + 2] cross-benzannulation of *β*-CF_3_-1,3-enynes with diynes [45] (Figure 8). Both *β*-F-1,3-enynes and *β*-perfluoroalkylated 1,3-enynes also are good reaction partners in these Pd-catalyzed [4 + 2] cross-benzannulation reaction affording corresponding trifluoromethylated and perfluoroalkylated benzene derivatives. This cycloaddition strategy proved to be effective for the rapid construction of aromatic fluorides from easily available acyclic starting materials.

In 2014, the Zhang group disclosed an iron-promoted electrophilic annulation of trifluoromethyl-containing aryl enynes with disulfides or diselenides affording polysubstituted naphthalenes in moderate to excellent yields. The reaction proceeded with high selectivity to provide the *6-endo-dig* cyclization product and showed good functional group tolerance. The authors observed that the aryl groups bearing electron-withdrawing substituents in these substrates results in lower yields. The utilization of inexpensive ferric chloride and commercially available disulfides and diselenides as electrophiles are significant advantages for the usefulness of this reaction [46] (Figure 9).

Based on their experimental results, a plausible mechanism was outlined, as shown in Figure 9b. The reaction of disulfide with iodine may take place to produce the active RSI in situ. The presence of BPO promotes the generation of free radical RS^•^ which subsequently reacted with I_2_ to form RSI. The electrophilic addition of RSI to the triple bond of enyne affords intermediate **A** (Path 1). The Lewis acid (FeCl_3_)-catalyzed intramolecular electrophilic attack on the neighboring aryl group provides intermediate **B**, and subsequent deprotonation yields the desired products. However, a free radical pathway cannot be ruled out (Path 2), in view of the fact that the product can be isolated in the absence of I_2_. The addition of RS^•^ to enyne produces a vinyl radical **C**, which then undergoes an intramolecular cyclization to form radical **D**. The following oxidation by I_2_ or FeCl_3_ produces the corresponding carbocation, which loses a H^+^ to yield desired products.

## 3. Construction of Trifluoromethylated Heterocycles

### 3.1. Construction of Five-Membered Trifluoromethylated Heterocycles

Recently, Hanamoto and co-workers developed a convenient method for the synthesis of 4-CF_3_-3-iodo-2-substituted thiophene from (*Z*)-2-bromo-2-CF_3_-vinyl benzyl sulfide in two steps. The Sonogashira cross-coupling reaction of (*Z*)-2-bromo-2-CF_3_-vinyl benzyl sulfide with various terminal acetylenes afforded the corresponding (*E*)-2-CF_3_-1-buten-3-ynyl benzyl sulfides in good to high yields. Subsequent iodocyclization afforded the corresponding 4-CF_3_-3-iodo-2-substituted thiophenes in good to high yields. It is noteworthy that the substrate bearing a triisopropylsilyl group was intact under optimal reaction conditions due to the bulky silyl group hindering the approach of electrophilic iodine [47] (Figure 10).

Other methods to construct thiophene derivatives from *β*-CF_3_-1,3-enynes were also reported. In 2020, Song group developed a divergent strategy for the construction of 3-SCF_2_H-4-CF_3_-thiophenes from readily available 1,3-enynes and S_8_ via a tandem thiophene construction/selective C3 thiolation/difluoromethylthiolation under a ClCF_2_H atmosphere with excellent substrate compatibility. Experiments had shown that the construction of the thiophene ring may be a radical annulation process with S_3_^•−^ generated in situ, and freon is used as a cheap difluoromethylation reagent. A series of 3-SeCF_2_H-4-CF_3_-selenophenes can also be constructed by similar strategies [48] (Figure 11).

In their subsequent work, they developed a divergent method for precise constructions of cyclic unsymmetrical diaryl disulfides or diselenides and polythiophenes from *β*-CF_3_-1,3-enynes and S_8_ when the ortho group is F, Cl, Br, and NO_2_ on aromatic rings. However, when the ortho group is H, disulfides (diselenides) were constructed These transformations undergo a cascade thiophene construction/selective C3-position thiolation process. A novel plausible radical annulation process was proposed and validated by DFT calculations [49] (Figure 12).

In 2014, we developed novel divergent cyclizations of *N*-(2-(trifluoromethyl)-3-alkynyl)hydroxylamines which are easily prepared from the corresponding *β*-CF_3_-1,3- enynes and the salt of hydroxylamine under basic reaction conditions via simple nucleophilic addition by subtle choice of the catalyst system, leading to two important trifluoromethylated nitrogen containing heterocycles, such as 4-trifluoromethyl cyclic nitrone and pyrrole. The IPrAuNTf_2_/HNTf_2_ co-catalyzed cyclization of *N*-(2-perfluoroalkyl-3-alkynyl) hydroxylamines produces pyrroles in moderate to excellent yields, whereas the AgOTf-catalyzed reaction affords cyclic nitrones in high yields. The notable features of the method are its easily accessible starting materials, mild reaction conditions and divergent synthesis [50] (Figure 13).

Following the above work, we subsequently developed a novel NIS-mediated oxidative cyclization of *N*-(2-trifluoromethyl-3-alkynyl) hydroxylamine under mild conditions, which provides a facile route to various 4-trifluoromethyl-5-acylisoxazoles. It was found that the NIS acts as both an oxidant and an electrophile for this sequential transformation. The key intermediate oxime formed by oxidation of hydroxylamine by NIS could be isolable, which underwent I^+^-induced *O*-selected *5-exo-dig* cyclization and subsequent cascade reaction to afford 4-trifluoromethyl-5-acylisoxazoles. It is also noteworthy that no desired product was obtained when alkyne bearing TMS group or terminal alkyne was used as substrate [51] (Figure 14a). Control experiments indicated that the oxygen atom of the ketone originated from water rather than from molecular dioxygen (Figure 14b).

The exclusive *O*-selected *5-exo-dig* cyclization of oxime in the above transformation conditions have aroused our interest because oximes can be employed as *N*-selective nucleophiles or as *O*-selective nucleophiles in many chemical transformations. We deduced that the Brønsted acid (HI) in situ formed in the above transformation conditions decreases the nucleophilicity of oxime nitrogen and destroys the inter- or intramolecular hydrogen bond between oxime oxygen and trifluoromethyl group, thus facilitating *O*-selective *5-exo-dig* cyclization. We envisioned that with a proper choice of transition metal catalysts, the isolated oximes might undergo *N*-selective *5-endo-dig* electrophilic cyclization owing to their inter- or intramolecular hydrogen bond between oxime oxygen and the trifluoromethyl group, which decreased the nucleophilicity of the oxime oxygen [52]. Thus, another type of interesting nitrogen containing heterocycle i.e., fluorinated *N*-hydroxypyrroles, could be obtained (Figure 15, path a). Furthermore, if excess NIS and molecular iodine (I_2_) formed in situ in the reaction were reduced with a proper reductant, the in situ formed Brønsted acid would catalyze the *O*-selective *5-exo-dig* or *6-endo-dig* electrophilic cyclization of oximes; thus, an easy two-step, one-pot-synthesis of 4-trifluoromethyl-5-alkylisoxazoles (Figure 15, path b) would be developed. Based on the above considerations, we developed the divergent regioselective cyclizations of *N*-(2-trifluoromethyl-3-alkynyl) oximes by subtle choice of gold(I) or Brønsted acid catalyst system, leading to 4-trifluoromethyl *N*-hydroxypyrroles or 5-akylisoxazoles. In order to avoid the tedious separation of unstable *N*-(2-trifluoromethyl-3-alkynyl) oximes, an easy two-step, one-pot synthesis of 4-trifluoromethyl-5-alkylisoxazoles from *N*-(2-trifluoromethyl-3-alkynyl) hydroxyl-amines is realized. This two-step, one-pot procedure is a complementary method for the synthesis of 4-trifluoromethyl-5-alkyl isoxazoles from those unstable *N*-(2-trifluoromethyl-3-alkynyl) oximes [53] (Figure 15).

After we studied the chemistry of *β*-CF_3_-1,3-enynes with the bisnucleophile hydroxylamine and subsequent transformations, we then studied the reaction of *β*-CF_3_-1,3-enynes with the bisnucleophile 2-aminomalonates. A dramatic substituent effect was observed in the reaction of *β*-CF_3_-1,3-enynes with the bisnucleophile 2-aminomalonates. When *N*-tosylated 2-aminomalonate was used as bisnucleophile, the reactions proceeded smoothly to afford 2-fluoro-2-pyrrolines via double direct C-F substitutions [54]. In contrast, either 4-trifluoromethyl pyrrolidines or *gem*-difluoro-1,3- conjugated enynes were delivered when *N*-acetylated 2-aminomalonate was used as reaction partner. *β*-CF_3_-1,3-enynes show an interesting substituent effect on the product diversity. *β*-CF_3_-1,3-enynes bearing electron-donating or weak electron-withdrawing groups, such as Me, MeO, Cl and Br, on the aryl substituent of the alkyne moiety afford functionalized gem-difluoro-1,3-conjugated enynes in moderate to good yields, whereas 4-trifluoromethyl pyrrolidines are isolated as the predominant product in moderate to good yields from those *β*-CF_3_-1,3-enynes with strong electron-deficient aromatic substituent. Various functionalized 4-(difluoromethylene)-1,2,3,4-tetrahydropyridines could be obtained in good yields via the gold(I)-catalyzed intramolecular *6-endo-dig* cyclization of the corresponding *gem*-difluoro-1,3-conjugated enynes under mild conditions [55] (Figure 16).

In 2017, we developed the first example of tandem intermolecular hydroamination and cyclization reaction of *β*-CF_3_-1,3-enynes with bisnucleophile primary amines affording 4-trifluoromethyl-3-pyrrolines by employing a cheap silver catalyst under mild reaction conditions. This new method is compatible with alkyl, aryl, and allyl primary amines, representing an atom-economic protocol for the construction of 4-trifluoromethyl-3-pyrrolines for the first time. It should be noted that the reaction also works well for aromatic primary amines, albeit requiring a higher reaction temperature [56] (Figure 17).

Following the above work, we subsequently developed a facile two-step, one-pot method for the synthesis of a range of halogenated trifluoromethylated pyrroles from *β*-CF_3_-1,3-enynes, readily aliphatic primary amines and halogenating agents, such as NBS and NIS. By variation of the halogenating agents, ring trifluoromethylated monoiodo pyrrole or dibromo pyrrole skeletons can be readily accessed in moderate to good yields. The different outcome of the reactions with NIS and NBS may be due to their different electrophilic properties towards pyrrole. This two-step, one-pot method employs a key halogenating-agents mediating step to trigger a cascade process featuring an initial electrophilic cyclization of the first intermolecular hydroamination product [57] (Figure 18).

More recently, in 2021, we developed the first example of a tandem double hydroamination reaction of *β*-CF_3_-1,3-enynes with bisnucleophiles hydrazine derivatives under mild reaction conditions. By variation of the substituents on the hydrazine nitrogen atom, three types of trifluoromethylated pyrazolidines, pyrazolines and pyrazoles can be readily accessed in moderate to good yields. The reaction with simple hydrazine monohydrate or sulfonyl hydrazines as nucleophiles produces 1,3,4-trisubstituted pyrazolines, whereas the reaction with acetyl hydrazine as nucleophiles affords 1,4,5-trisubstituted pyrazolidines. Using phenylhydrazine or tert-butylhydrazine as reaction partners, the products are easily oxidized to form 1,4,5-trisubstituted pyrazoles [58] (Figure 19).

### 3.2. Construction of Six-Membered Trifluoromethylated Heterocycles

In 2000, the Qing group first reported the synthesis of 4-trifluoromethyl-2*H*-pyrans by palladium-catalyzed cyclization of (*E*)-3-alkynyl-3-trifluoromethyl allylic alcohols. The *6-endo-dig* cyclization, not the *5-exo-dig* cyclization, is favored due to the fact that the trifluoromethyl group possesses powerful electron-withdrawing properties and (*E*)-3-alkynyl-3-trifluoromethyl allylic alcohols bearing an aryl group adjacent to the hydroxyl lead to dienone under same reaction conditions [59] (Figure 20).

A straightforward and efficient approach to alkyne-functionalized ring-monofluorinated 4*H*-pyrans via a simple base-mediated cascade reaction of *β*-CF_3_-1,3-enynes with 1,3-dicarbonyl compounds or monocyano- substituted carbon nucleophiles, such as 3-oxo-3-phenylpropanenitrile, 3-oxo-butyronitrile was developed by our group [60] and Chang [10]. Substituted alkynyl group was used as an activating group. The key events of this reaction involve two consecutive C-F substitutions under very mild conditions (Figure 21).

Very recently, we developed the first example of a tandem intermolecular hydrocarbonation/intramolecular heterocyclization reaction of *β*-CF_3_-1,3-enynes with bisnucleophiles *β* -ketothioamides under mild reaction conditions. By variation of the substituents linked to the carbonyl or on the *β*-ketothioamides nitrogen atom, ring trifluoromethylated pyrans, or thiopyrans, can be readily accessed in moderate to good yields. Enynes possessing electron-withdrawing aryl groups on the alkyne moiety are generally good candidates for present transformation and *β*-ketothioamides bearing a piperidine substituent on the amide moiety, and (hetero)aryl groups on the keto moiety would mainly afford pyran, whereas *β*-ketothioamides bearing pyrrolidine substituent on the amide moiety and (hetero)aryl or alkyl groups on keto moiety lead to the formation of thiopyrans. Other substituted forms of *N*,*N*-disubstituted *β*-ketothioamides would give mixtures of pyrans and thiopyrans. We think that the formation of the oxygen enol or sulfur enol intermediate **Int-B** could be affected by electronic and spatial effects of substituents on either the keto moiety or the nitrogen atom of the *β*-ketothioamide from the reaction of **Int-A** under basic conditions, which results in pyrans or thiopyrans. *β*-ketothioamides bearing a piperidine substituent on the amide moiety and (hetero)aryl groups on the keto moiety would mainly form oxyenols, leading to the formation of pyran, whereas sulfur enol would predominantly form *β*-ketothioamides bearing a pyrrolidine substituent on the amide moiety and (hetero)aryl or alkyl groups on the keto moiety, leading to the formation of thiopyrans. Other substituted forms of *N*,*N*-disubstituted *β*-ketothioamides would give mixtures of oxygen enol and sulfur enol which result in mixtures of pyrans and thiopyrans. The salient features of this tandem include atom-economical, mild reaction condition, ease of operation and product diversity [61] (Figure 22).

## 4. Construction of Other Value-Added Trifluoromethylated or Non-Fluorinated Organic Compounds

The divergent synthesis of thioether-functionalized trifluoromethyl-alkynes, 1,3-dienes and allenes from the regioselective nucleophilic addition reactions of *β*-CF_3_-1,3-enynes with sulfur nucleophiles was discovered by our group in 2018. The addition patterns depend on the type of enynes, sulfur nucleophiles and reaction conditions used. 1,4-addition leading to thioether-functionalized trifluoromethyl-allenes was realized when enynes possessing electron-withdrawing aryl groups on the alkyne moiety were used as reaction partners and alkanethiols were used as nucleophiles, whereas solvent-controlled construction of thioether-functionalized 1,3-dienes and alkynes were realized, respectively, via 3,4-addition pattern or 1,2-addition pattern if thiophenols were applied as nucleophiles. The three types of compounds containing both sulfur and fluorine elements are valuable building blocks for synthesis of multifunctional trifluoromethylated vinyl sulfides and thiophenes derivatives [62] (Figure 23).

In 2020, Song and coworkers reported three unprecedented Cu-catalyzed regio- and stereo-divergent chemoselective sp^2^/sp^3^ 1,3- and 1,4-diborylation of *β*-CF_3_-1,3-enynes, affording a broad array of diborylated compounds containing CF_3_ group in simple and efficient ways. Homopropargylic boronates and homoallenyl boronates as the key intermediates for the above three transformations were obtained after carefully modifying the reaction conditions. DFT calculations explain the reactivity, regioselectivity, as well as the stereoselectivity in these transformations in detail [63] (Figure 24).

In the same year, a regio-divergent boroprotonation of *β*-CF_3_-1,3-enynes controlled by ligand for divergent synthesis of CF_3_-substituted homopropargyl boronates and homoallenyl boronates was reported by Cao and coworkers They found that XantPhos promoted the 1,2-boroprotonation of *β*-CF_3_-1,3-enynes in good yields and with complete regiocontrol. Conversely, 1,4-boroprotonation products were accessed with high selectivity and in moderate yield upon switching to the bidentate, nitrogen-based ligand 4,4′-di-tert-butyl-2,2′-bipyridine (dtbpy) [64] (Figure 25).

In their subsequent work, base-catalyzed nucleophilic additions of TMSCN to *β*-CF_3_-1,3-enynes, affording 1,2-hydrocyanation or 1,4-hydrocyanation products in good to excellent yields with high regioselectivity, were developed. When the reaction of *β*-CF_3_-1,3-enynes with TMSCN was carried out in the presence of 20 mol% DBU, the reactions afforded the 1,4-hydrocyanation products i.e., cyanated CF_3_-substituted 1,3- butadienes in fair to excellent yields, whereas the reaction provided the 1,2-hydrocyanation products by using 5 mol% Cs_2_CO_3_ instead of DBU [65] (Figure 26).

The Cu-catalyzed 1,4-protosilylation and protoborylation of *β*-CF_3_-1,3-enynes to access functionalized homoallenyl silanes and homoallenyl boronates were developed by Xu and co-workers. This protocol also provides a general method to synthesize optically active homoallenyl silanes and homoallenyl boronates in moderate to excellent yields with high enantiomeric excess by using new designed chiral bisoxazoline ligands [66] (Figure 27).

In their subsequent work, an interesting selective and diverse defluoromethoxylation reactions of *β*-CF_3_-1,3-enynes with potassium methoxide affording enynic and allenyl orthoesters under mild reaction conditions was developed. The transformations of enynic orthoesters proved that this class of compounds are efficient and flexible “platform molecules” for the synthesis of various functionalized allenes [67] (Figure 28).

Recently, Liu and coworkers reported the first 1,2-dicarbofunctionalization of *β*-CF_3_-1,3-enynes with pyridinium salts via a cascade process involving a base-promoted [3 + 2] cycloaddition followed by a visible-light-mediated Norrish-type-II fragmentation. This protocol allows for the formation of pyridines bearing a trifluoromethyl-substituted quaternary center in moderate to excellent yields under mild conditions. Besides *β*-CF_3_-1,3-enynes, other trifluoromethyl alkenes, such as α-CF_3_-(hetero)aryl alkenes, and *N*-[(α-trifluoromethyl)vinyl]imides, are good candidates for this transformation [68] (Figure 29).

## 5. Conclusions

In this review, we have summarized recent efforts to develop new diversity-oriented synthesis based on *β*-CF_3_-1,3-enynes in one type of fluorine-containing building blocks. The distinct reactivity of *β*-CF_3_-1,3-enynes and their variants allow these reactions to deliver numerous value-added fluorine-containing compounds which have biologically relevant structural motifs, such as *O*-, *N*-, and *S*-heterocycles, carboncycles, fused polycycles, and multifunctionalized allenes. While the advances made to date are remarkable, further inventing novel and efficient transformations of *β*-CF_3_-1,3-enynes to synthesize diverse trifluoromethylated cyclic molecules, such as furan and acyclic molecules, is still interesting and urgent.

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
