# Peer review of "Versatile Fluorine-Containing Building Blocks: β-CF3-1,3-enynes"

_molecules, 2022, doi:10.3390/molecules27249020_

Round 1

Reviewer 1 Report

In general, this is a welcome review on applications of CF3-substituted 1,3-enynes as building blocks for the construction of 3-, 5-, and 6-membered carbocyclic and different heterocyclic compounds by different kinds of cycloaddition and heterocyclization reactions.

General comments

For almost all syntheses shown in the Schemes, a mechanistic explanation for the pictured transformation(s) is missing. In my opinion, a more scholarly presentation instead of a simple enumeration of the results would greatly enhance the value of the manuscript. Moreover, there are numerous errors and inaccuracy in the text, e.g. nomenclature issues.

Specific comments

-      The literature cited in ref. [1] is not up to date. There are important more recent reviews and books on the topic. The authors might want to add literature published within the past 5 years.

-      The same is true for the literature cited in ref. [2]. To me the selection seems random.

-      A better linkage of this quite general first part to the specific topic of the review is recommended.

-      Lines 8, 14, 16, 24, 35, …throughout the whole manuscript: … “β-CF3-1,3-Enynes” and similar cases should read “β-CF3-1,3-enynes” (no capital characters within a term of nomenclature)

-      Line 29: after “… the building blocks” a reference (preferably a review) is expected

-      There is an error in scheme 2 regarding the solvent(s) on the arrow

-      Line 78: it should read “The Trost group …” (or Trost’s group …)

-      Line 79: … α-trifluoromethyl … (not a capital T), and many similar cases throughout the whole text

-      The nomenclature should be checked throughout the manuscript (by way of example: lines 152 and 153 … “(Z)-2 bromo-2-CF3- vinyl benzyl …” should read: “(Z)-2-bromo-2-CF3-vinyl benzyl …” (extra spaces and hyphenation). This is only one example. There are many more of similar cases.

-      Particularly in Schemes 13 and 14, a mechanistic explanation is missing. In the latter one, the role of I+ is not clear and it is not mentioned where the oxygen of the keto group is coming from (water is not mentioned in the scheme)

-      It is confusing to use the same code number(s) for different molecules in different Schemes, and even within the same Scheme (Scheme 14)

-      Line 213: … as N- selective (delete the extra space). There are many similar cases in the manuscript

-      Lines 219 and 226: … 3-alkynyl) oximes …

-      Line 223: … hydroxyl-amines … (not hydroxyla-mines), and many similar cases

-      The difference between the I+-induced reactions shown in scheme 14 and the H+-induced reactions in scheme 15 leading either to ketones or hydrocarbons (in this specific position) should be explained in the text

-      Line 262: … we subsequently developed …

-      An explanation for the different outcome of the reactions with NIS and NBS (Scheme 18) should be given

-      Scheme 21 is confusing, there is no 1,3-dicarbonyl compound involved in the shown reaction scheme

-      Please add a verbal explanation for the different reaction mode of pyrrolidine or piperidine-substituted compounds in Scheme 22

In summary, I recommend major revision and resubmission of the manuscript.

Reviewer 2 Report

Development of methods aimed at synthesis of carbo- and heterocyclic compounds containing fluorine or fluoroalkylated groups is of current interest. The review submitted by Xiao et al. summarizes results related to exploration of beta-CF3 functionalized 1,3-enynes for construction of carbo- and heterocyclic compounds. The review is based on 34 references (some of them with multiple publications) from recent two decades.  The review fits well in the profile of Molecules and is focused on relevant questions of the current chemistry of fluorine organic compounds. In my opinion it can be recommended  for publication in the Journal after minor revision.

Following remarks are addressed to Authors in the course of preparation of the revised version:

1)      To the list of publications cited at ref. [2]: a very recent review on methods related to synthesis of fluorinated and fluoroalkylated sulfur heterocycles has to be added. Title: Synthesis of fluorinated and fluoroalkylated heterocycles containing at least one sulfur atom. Materials 2022, 15, 7244,

2)      Captions for all schemes should be given in format of Molecules, e.g. Scheme 1. Synthesis of trifluoromethylated cyclopropanes …

In general: ‘fluorinated’ should be replaced by ‘trifluoromethylated’ or ‘difluoromethylidene’ in many schemes and also in the main text; there is a clear difference between ‘fluorinated’ and ‘fluoroalkylated’!

3)      In Scheme 2: in the equation, the same solvent (DCE) is given for products from left and from right. This is not true (see the comment given above in lines 58-60).

4   Chapter 3: the title should be modified (see my comment #2).

In summary: after careful revision, the manuscript can be recommended for publication in Molecules.   

Round 2

Reviewer 1 Report

no additional comments